# General Self-Efficacy as a Mediator of Physical Activity’s Impact on Well-Being Among Norwegian Adolescents: A Gender and Age Perspective

**DOI:** 10.3390/bs15091239

**Published:** 2025-09-11

**Authors:** Karianne Fossli, Catherine A. N. Lorentzen

**Affiliations:** Department of Health, Social and Welfare Studies, University of South-Eastern Norway, P.O. Box 4, 3199 Borre, Norway; catherine.lorentzen@usn.no

**Keywords:** physical activity, subjective well-being, general self-efficacy, adolescents, gender, age, mediator, moderator

## Abstract

Background: The mechanisms behind the beneficial effects of physical activity on adolescent well-being remain poorly understood. One potential pathway involves increased general self-efficacy. A deeper understanding of underlying processes, and of direct and indirect impacts across adolescent sub-groups, can improve intervention strategies. This study investigates whether physical activity predicts subjective well-being in adolescents, the mediating role of general self-efficacy, and the moderating effect of gender and age on these relationships. Methods: This cross-sectional survey study utilised data from a population-based sample of 18,146 Norwegian adolescents aged 14–19. Simple and moderated mediation models were tested with Hayes’ PROCESS in SPSS, with life satisfaction as the dependent variable, physical activity as the independent variable, general self-efficacy as the mediator, and gender and age as moderators. Analyses were controlled for socio-demographic variables and social support. Results: The findings indicated a small, positive effect of physical activity on well-being, consistent across genders but increasing with age, though not significant for those under 14.5 years. Approximately half of the total effect of physical activity on well-being was mediated through general self-efficacy. This indirect effect was greater for girls and younger adolescents compared to their counterparts, due to greater positive impacts of general self-efficacy on well-being within these two sub-groups. Conclusions: By identifying general self-efficacy as a mediator of physical activity’s impact on adolescent well-being, along with distinct direct and indirect effects within sub-groups, this study enhances the theoretical framework and knowledge base for more effective physical activity initiatives targeting the well-being of this demographic.

## 1. Introduction

A trend of declining mental health has been observed among adolescents in Western societies over the past 10–15 years ([8]; [57]; [77]; [89]; [102]). This encompasses both the negative dimension of mental health, such as psychological distress and mental illness, and the positive dimension, including perceptions of well-being ([56]; [61]). This situation presents significant challenges for the affected individuals, their family and friends, and society at large, both currently and in the future ([78]; [119], [120]), and underscores the urgent need to deepen our understanding of factors that might mitigate these effects ([120]). Various biological, behavioural, social, cultural and structural factors have been documented to influence adolescents’ mental health ([8]; [118], [119]). Among these, physical activity stands out as a significant contributor across a range of negative and positive mental health outcomes ([9]; [17]; [26]; [47]; [67]; [70]; [94]; [116]).

However, while the beneficial mental health impact of physical activity is well established, the mechanisms explaining *how* this occurs are less well understood ([9]; [26]; [67]; [94]; [110]; [116]). A recent systematic review by [116] ([116]), covering studies across diverse population groups, provides the most comprehensive synthesis to date of potential mediators underlying this effect. The review identifies several key mediating factors supported by strong evidence, e.g., affect, self-esteem, and social support. [67] ([67]) have proposed a conceptual model to organise mediators of these relationships, grouping them into three distinct domains: neurobiological, behavioural and psychosocial.

Neurobiological mechanisms involve changes in brain structures and functions, including enhanced cognitive function related to attention and working memory—factors important for maintaining good mental health ([41]; [67]; [117]). Additionally, physical activity promotes the release of neurotransmitters like endorphins, dopamine, noradrenaline and serotonin, which positively influence mood, enhance feelings of happiness and alleviate stress ([30]; [58]; [67]; [122]). Proposed behavioural mechanisms include improvements in self-regulation skills ([67]), sleep patterns ([58]; [67]), and other lifestyle behaviours such as substance use, diet, screen-time and academic behaviours ([67]). Psychosocial mechanisms include enhanced physical self-perception ([9]; [26]; [67]), self-esteem ([26]; [63]; [67]; [116]), general self-efficacy ([116]), and social support ([21]; [34]; [116]). A sense of independence and autonomy has also been suggested as a psychosocial mechanism ([67]).

While the psychosocial pathway through enhanced general self-efficacy was proposed more than two decades ago ([87]), and is generally well-supported across diverse populations ([116]), this mechanism has received little empirical attention in adolescent populations ([67]; [94]; [116]). General self-efficacy (GSE) refers to a broad, stable belief in one’s ability to manage various difficult or challenging life situations ([5]; [98]; [100]; [115]). An expanding array of cross-sectional and longitudinal research has revealed positive links between adolescents’ perceptions of GSE and a range of mental health outcomes, both negative ([12]; [54]) and positive (e.g., [64]; [16]; [44]; [73]; [75]; [103]) ones. This effect may be explained by several processes. Firstly, optimistic beliefs about one’s competence and ability to handle future challenges enhance positive affect ([68]). Secondly, individuals with higher GSE tend to be more engaged and dedicated across various activities, as they are more likely to view difficult tasks and situations as opportunities for mastery, set ambitious goals and persist in achieving them ([5]; [85]; [100]). This often results in greater enjoyment and higher life satisfaction. Thirdly, in line with the latter, individuals with high GSE are more inclined to engage in behaviours that inherently promote mental health. For example, they are more likely to participate in social activities, which may offer additional psychological benefits ([10]; [68]). Finally, high GSE may buffer against the negative effects of adverse experiences and critical feedback from the environment ([18]).

While not extensively studied, several investigations have demonstrated a positive association between physical activity or sports participation and GSE in adolescents ([65]; [40]; [43]; [76]; [91]; [93]; [94]; [111]; [124]). Theory and research point to three primary sources of self-efficacy: personal experiences of mastery, as they offer insights into one’s capabilities and help shape beliefs regarding future achievements; vicarious experiences of mastery, i.e., where observing social role models successfully tackling challenges reinforces one’s own self-efficacy; and social persuasion, which involves significant others verbally affirming one’s capabilities ([6], [7]; [85]; [108]). It is likely that a wider range of physical activity behaviours and higher levels of physical activity can provide access to more of these self-efficacy-enhancing experiences. For example, mastery can be achieved by improving technical skills specific to the activity or by developing general skills such as setting goals, planning, and executing plans ([59]). Additionally, engaging in physical activities alongside others, such as in sports clubs and fitness centres, can provide access to successful role models and encouraging social environments ([92]).

However, few studies have investigated the entire chain of effects between physical activity, GSE and mental health in adolescent populations ([31]; [53]). While existing cross-sectional and longitudinal studies among adults ([28]; [46]; [79]; [114]; [124]) all support the mediating role of GSE in these associations, the only two studies that have investigated this topic in adolescent populations have shown mixed results. The cross-sectional study by [31] ([31]) involving 1491 Chinese adolescents aged 12–18 also supported GSE as a mediator, while [53]’s ([53]) cross-sectional study among 775 Chinese adolescents aged 12–14 did not.

Collectively, the reviewed literature indicates that GSE may play a role in explaining better mental health among adolescents with higher levels of physical activity. Investigating this hypothesis, preferably in a large, population-based sample of adolescents, would contribute to theoretical advancements in the field and inform public health policy and practice, facilitating the design of more effective physical activity-promoting initiatives aimed at improving the mental health of this target population.

Moreover, studies continuously show that adolescent girls in Western societies report poorer mental health ([3]; [25]; [102]), lower GSE ([64]; [10]; [55]; [66]; [75]), and lower physical activity levels ([2]; [101]; [104]) compared to boys. Furthermore, studies reveal poorer mental health ([15]; [49]; [84]), lower physical activity levels ([2]; [101]), but stronger GSE among older adolescents compared to the younger ones ([43]). A few studies have investigated whether gender or age moderates associations between physical activity, GSE and mental health. In an umbrella systematic review, [26] ([26]) found mixed results as to whether the impact of physical activity on mental health among adolescents was dependent on gender or age. [111]’s ([111]) cross-sectional study indicated that the association between physical activity and GSE was stronger for girls than boys. However, [65] ([65]) did not find evidence of such gender interaction effects, nor did they observe age-related interaction effects. Two studies found that the predictive effect of GSE on mental health was more pronounced in adolescent girls than boys ([64]; [12]), while another study did not identify such a gender difference, nor an age difference ([49]). [53] ([53]), in their study on the mediating role of GSE in the association between physical activity and mental health among Chinese adolescents, found the same non-significant indirect effect for both boys and girls. Despite showing mixed results, current research suggests potential variations in the direct and indirect effects (via GSE) of physical activity on mental health outcomes across gender and age sub-groups within the adolescent population. Exploring these aspects is necessary to further develop theory in this area and establish a foundation for better-tailored interventions ([9]; [22]; [48]).

Finally, research suggests that negative and positive mental health outcomes, while sharing some commonalities, also represent distinct aspects of mental health ([90]). For instance, an individual may live with mental illness yet still experience a sense of well-being, whereas someone without a mental illness may not necessarily feel content or satisfied with their life. There have been increasing calls for more research into the factors contributing to the less studied positive mental health outcomes, such as subjective well-being (SWB) ([90]).

### Aim of the Study

Addressing gaps in existing research, this study aims to examine (1) whether physical activity predicts SWB in a large population-based sample of Norwegian adolescents, and whether GSE contributes as a mediator of this effect, and (2) whether gender and age moderate direct and indirect effects of physical activity on SWB.

## 2. Materials and Methods

### 2.1. Study Design, Data Collection, and Sample

This cross-sectional study is based on self-reported survey data from the Norwegian Ungdata programme ([39]), which was designed to gather public health-relevant data from adolescents in lower and upper secondary schools across Norwegian counties. Data collection is conducted electronically during school hours, typically every three years in schools within a given county. The data for this study originate from the 2021 collection conducted in Vestfold and Telemark County. Of the 27,833 invited pupils, 22,028 responded, resulting in an 86% response rate for lower secondary pupils and 72% for upper secondary pupils ([1]).

The analytical sample for this study comprises the 18,146 pupils who completed the items required to construct the variables included in the main analyses of the study—the mediation and moderated mediation analyses. Attrition analyses showed that participants included in the analyses reported slightly higher levels of physical activity, GSE, SWB, family socio-economic status, and social support compared to those excluded due to missing data (see Table 1). They were also slightly older and more likely to be girls.

### 2.2. Ethics

Participation in the study was based on voluntary involvement and informed consent, with passive consent from parents/guardians of pupils under 18 years of age, meaning consent was provided by not opting out their adolescent after receiving study information ([82]). The data used were already anonymised and accessed through the [80] ([80]).

### 2.3. Measurements

#### 2.3.1. Subjective Well-Being

The dependent variable, SWB, was assessed through one item capturing adolescents’ overall evaluation of life satisfaction, a cognitive sub-dimension of SWB ([78]): “Overall, where do you place yourself on this scale at the moment?” The scale ranged from 0 (worst possible life) to 10 (best possible life). Single-item measurements of SWB are commonly employed in large-scale surveys with restricted questionnaire space ([78]), and they are found to be valid and reliable ([29]; [62]).

#### 2.3.2. Physical Activity

The independent variable, physical activity, was measured by the following item: “How often are you physically active to the point of becoming breathless or sweaty?”. Response categories included: “Never” = 1, “Rarely” = 2, “Once to twice a month” = 3, “Once to twice a week” = 4, “Three to four times a week” = 5, “At least five times a week” = 6. This ordinal variable was treated as continuous.

#### 2.3.3. General Self-Efficacy

The mediating variable, GSE, was assessed by the Norwegian short version of the General Perceived Self-Efficacy Scale ([95]; [99]), which comprises five statements: “I always manage to solve difficult problems if I try hard enough”, “I feel confident that I would be able to deal with unexpected events in an effective way”, “I remain calm when I face difficulties because I trust my ability to cope”, “If someone opposes me, I can find the means and ways to get what I want”, “If I’m in a predicament, I usually find a way out”. Response options included: “Completely wrong” = 1, “Quite wrong” = 2, “Quite true” = 3, and “Completely true” = 4. A mean score was computed for respondents who had answered at least three out of the five statements, creating a variable ranging from 1 to 4, where higher scores denoted greater GSE. Both the original ten-item scale and the Norwegian short version have demonstrated good psychometric properties across various populations, including adolescents ([66]; [96]; [97]; [100]; [105]; [123]). The short version also exhibited a high level of internal consistency in this study’s sample (α = 0.87).

#### 2.3.4. Moderator and Control Variables

Gender was assessed with the categories Boy (=0) and Girl (=1). Respondents’ age was determined by year level (first year of lower secondary school = 14 years, last year of upper secondary school = 19 years). This variable was dichotomised for some initial analyses (14–16 years and 17–19 years) but treated as continuous in the main analyses. Family socio-economic status was assessed by aggregating responses to six items reflecting the family’s economic, cultural, and social resources ([4]). Four items were from the Family Affluence Scale II, assessing wealth status via questions on (1) car and (2) computer ownership, (3) holiday travel, and (4) whether the adolescent has their own bedroom ([24]). Another item measured parents’ educational level, and the last item inquired about the number of books at home. Following the guidelines from the developers of the Ungdata questionnaire, responses for these six items were coded from 0 (low socio-economic status) to 3 (high socio-economic status) ([4]). For instance, the parental educational scale included the following categories: “None of the parents have higher education” = 0, “One of the parents” = 1.5, and “Both parents” = 3. A mean score of these six scales was computed. Social support was assessed with the item: “If you feel down or sad and need someone to talk to, do you have someone to talk to?”. Response categories were (recoded): “No” = 0, “Don’t know” = 1, and “Yes” =2. This variable was mainly treated as categorical.

### 2.4. Statistical Analyses

All statistical analyses were conducted using IBM SPSS Statistics, version 29 (IBM Corp., Armonk, NY, USA). Attrition analyses included *t*-tests to evaluate differences in means for continuous variables and Chi-square tests to assess differences in proportions for categorical variables between included and excluded respondents. Sample characteristics were assessed through means and standard deviations for continuous variables and frequencies and percentages for categorical variables, both for the entire sample and segmented by gender and age groups. *T*-tests and Chi-square tests were conducted to assess differences between sample sub-groups.

Bivariate associations among all study variables were assessed using correlation analyses, both for the entire sample and segmented by gender and age groups. Pearson correlation coefficients (r) were used for analyses involving continuous and dichotomous variables, while Spearman Rank Order correlation coefficients (rho) were applied in analyses including ordinal variables ([86]). Correlation strength was interpreted according to [19]’s ([19]) guidelines: 0.10–0.29 = small, 0.30–0.49 = medium, and 0.50–1.00 = large.

The study’s main analyses, comprising a mediation analysis and two moderated mediation analyses, were conducted using the PROCESS application 4.2 in SPSS ([50]). Hayes’ simple mediation Model 1 was employed to address the study’s first aim—whether physical activity (X) predicts SWB (Y), and if this effect is mediated by GSE (M) (see Figure 1a).

The mediation analysis assessed the total (c), direct (c’) and indirect (ab) effects (through GSE) of physical activity on SWB. The analysis controlled for gender, age, family socio-economic status, and social support. The study’s second aim—to determine whether the direct and indirect paths (via GSE) of an effect of physical activity on SWB are moderated by gender or age—was examined using Hayes’ moderated mediation Model 59, controlling for the remaining control variables. This was executed twice, once with gender and once with age as the moderator. These analyses evaluated the moderating impact of gender and age on each path (a, b, c, and c’) in the prior mediation analysis, as well as on the indirect path (ab) (see Figure 1b). Simple slope analyses were conducted to facilitate the interpretation of significant interaction effects. Unstandardised regression coefficients were assessed from these mediation and moderated mediation analyses. Tests were considered statistically significant if the 95% confidence interval excluded 0 ([51]). Confidence intervals for the indirect effects of physical activity on SWB were bootstrapped using 5000 resamples. All models are based on linear regression analyses, thus assumptions for linear regression—linearity, normality, non-collinearity, and homoscedasticity—were tested and found to be met ([37]; [86]). Missing data was handled via listwise deletion ([51]).

## 3. Results

### 3.1. Sample Characteristics

The sample characteristics are shown in Table 2, for the entire group and separately for girls and boys and for lower and upper secondary school pupils.

The sample had a slightly higher proportion of girls (51.6%) compared to boys (48.4%). The mean age was 16.3 years (*SD* = 1.61). Respondents reported relatively high mean SWB, with boys scoring higher than girls (*M* = 7.52, *SD* = 1.78 vs. *M* = 6.71, *SD* = 1.96, M diff. = 0.81, 95% CI M diff. = 0.76; 0.87, scale = 0–10) and younger respondents scoring slightly higher than older ones (*M* = 7.16, *SD* = 1.97 vs. *M* = 7.03, *SD* = 1.86, M diff. = 0.13, 95% CI M diff. = 0.07; 0.18). Respondents also reported relatively high GSE, with boys scoring higher than girls (*M* = 3.07, *SD* = 0.61 vs. *M* = 2.79, *SD* = 0.59, M diff. = 0.28, 95% CI M diff. = 0.26; 0.29, scale = 1–4), and older respondents scoring slightly higher than younger ones (*M* = 2.97, *SD* = 0.60 vs. *M* = 2.89, *SD* = 0.63, M diff. = −0.08, 95% CI M diff. = −0.09; −0.05). The average physical activity level ranged between “once to twice a week” and “three to four times a week”, with boys being more active than girls (*M* = 4.69, *SD* = 1.24 vs. *M* = 4.40, *SD* = 1.22, M diff. = 0.29, 95% CI M diff. = 0.25; 0.33, scale = 1–6), and younger respondents slightly more active than older ones (*M* = 4.59, *SD* = 1.21 vs. *M* = 4.48, *SD* = 1.26, M diff. = 0.11, 95% CI M diff. = 0.08; 0.15).

### 3.2. Correlation Results

Table 3, Table 4 and Table 5 display the results from correlation analyses among all study variables for the total sample and segmented by gender and age groups, respectively.

In line with [19]’s ([19]) guidelines, physical activity showed a small, positive correlation with SWB for both boys (r = 0.12) and girls (r = 0.15), as well as younger (r = 0.13) and older (r = 0.18) adolescents. Similarly, physical activity had a small, positive correlation with GSE for boys (r = 0.19), girls (r = 0.18), younger (r = 0.19), and older (r = 0.22) adolescents. GSE correlated positively with SWB, exhibiting a small effect size among boys (r = 0.25), a medium effect size among girls (r = 0.38), and medium effects among both younger (r = 0.38) and older (r = 0.31) adolescents.

### 3.3. Mediation Analysis (Aim 1)

Figure 2 presents the results from the mediation analysis.

There was a small, positive significant total effect of physical activity on SWB (path c: b = 0.13, SE = 0.01, 95% CI = 0.11; 0.15). Approximately half of this effect was direct (path c’: b = 0.07, SE = 0.01, 95% CI = 0.05; 0.10) while the remainder was mediated through GSE (path ab—indirect effect: b = 0.06, bootstrapped SE = 0.00, bootstrapped 95% CI = 0.05; 0.07). This was based on a small positive predictive effect of physical activity on GSE (path a: b = 0.07, SE = 0.00, 95% CI = 0.07; 0.08) and a medium-sized positive effect of GSE on SWB (path b: b = 0.79, SE = 0.02, 95% CI = 0.75; 0.83).

### 3.4. Moderated Mediation Analyses (Aim 2)

Figure 3 illustrates the results of interaction analyses testing gender as a moderator for all paths in the mediation model.

The analyses revealed that neither the direct effect of physical activity on SWB (path c’) nor the effect of physical activity on GSE (path a) depended on gender. However, the effect of GSE on SWB (path b) was moderated by gender (Interaction effect: b = 0.44, SE = 0.04, 95% CI = 0.36; 0.52). Further exploration of this interaction effect revealed a significant positive effect of GSE on SWB for both genders, but the effect was approximately twice as large for girls compared to boys (Girls: b = 1.01, SE = 0.03, 95% CI = 0.95;1.07, Boys: b = 0.57, SE = 0.03, 95% CI = 0.51; 0.63). This difference is depicted in the simple slope in Figure 4.

Due to this moderating effect, the indirect effect of physical activity on SWB through GSE (path ab) was significantly greater for girls than for boys (Boys: b = 0.04, bootstrapped SE = 0.00, 95% bootstrapped CI = 0.04; 0.05, Girls: b = 0.07, bootstrapped SE = 0.01, 95% bootstrapped CI = 0.06; 0.08).

Equivalent analyses with age as the moderator revealed a small moderating effect on the direct effect of physical activity on SWB (path c’) (Interaction effect: b = 0.02, SE = 0.01, 95% CI = 0.01; 0.03, see Figure 5).

Further exploration of this interaction effect showed that the positive effect of physical activity on SWB increased steadily with age but was non-significant for those under 14.5 years (14 years: b = 0.03, SE = 0.02, 95% CI = −0.01; 0.06, 16 years: b = 0.07, SE = 0.01, 95% CI = 0.05; 0.09, 18 years: b = 0.11, SE = 0.02, 95% CI = 0.08; 0.14). This interaction is depicted in Figure 6.

The effect of physical activity on GSE (path a) was not moderated by age. The effect of GSE on SWB (path b) showed slight moderation by age (Interaction effect: b = −0.06, SE = 0.01, 95% CI = −0.08;−0.03). GSE positively predicted SWB across all ages, but the effect diminished with increasing age (14 years: b = 0.91, SE = 0.04, 95% CI = 0.84; 0.98, 16 years: b = 0.80, SE = 0.02, 95% CI = 0.76; 0.84, 18 years: b = 0.69, SE = 0.03, 95% CI = 0.63; 0.75). This interaction is depicted in Figure 7.

Consequently, the indirect effect of physical activity on SWB via GSE (path ab) decreased slightly with age (14 years: b = 0.07, SE = 0.01, 95% bootstrapped CI = 0.05; 0.08, 16 years: b = 0.06, SE = 0.00, 95% bootstrapped CI = 0.05; 0.07; 18 years: b = 0.05, SE = 0.00, 95% bootstrapped CI = 0.04; 0.06).

## 4. Discussion

This study sought to deepen our understanding of the mechanisms through which physical activity enhances SWB in adolescents and to assess whether direct and indirect effects vary among sub-groups. We identified a positive effect of physical activity on SWB, consistent across genders and increasing with age, though non-significant in the very youngest participants. About half of the total effect of physical activity on SWB was mediated by GSE, with this indirect effect being more pronounced among girls and younger adolescents.

### 4.1. Direct Effects of Physical Activity on SWB

The mental health benefits of physical activity observed in this sample align results from a myriad of previous studies ([9]; [17]; [26]; [34]; [47]; [67]; [70]; [94]; [121]). Consistent with earlier research, the effect size is considered small. However, [94] ([94]) suggest that large effects may not be expected, as most young people already experience good mental health levels, despite the overall diminishing trend ([8]; [77]; [89]; [102]). Nevertheless, even small changes in large population groups are expected to significantly impact public health ([14]; [71]). Of particular interest in our study is the moderating role of age on the direct effect of physical activity on SWB. Previous research has yielded mixed results on this issue ([26]). For instance, [45] ([45]) revealed similar positive associations between physical activity levels and life satisfaction among lower and upper secondary pupils, but an inverse relationship with psychological distress only in the older group. The stronger relationships observed with increasing age in our study may be attributed to the consistently reported poorer quality of life among older adolescents compared to their younger counterparts ([15]; [49]; [84]; [112]), also evident in our sample. This suggests a greater potential for mental health improvement through increased physical activity levels in older adolescents. It is possible that the impact of physical activity on previously highlighted psychosocial and behavioural factors important for good mental health, such as physical self-perception and sleep patterns ([67]), are particularly important for older adolescents. For instance, studies indicate that dissatisfaction with physical appearance ([72]) and poor sleep behaviour ([3]) are more prevalent among older compared to younger adolescents.

### 4.2. Indirect Effect—Through GSE—Of Physical Activity on SWB

The primary aim of this study was to examine whether GSE mediates an association between physical activity and SWB. Our results confirmed the significance of adolescents’ general sense of ability to tackle upcoming challenging situations as a contributing underlying pathway in the mental health-enhancing effects of physical activity. Our findings show that the indirect effect occurred due to a small positive impact of physical activity on GSE beliefs, which subsequently exerted a medium-sized positive effect on well-being. These results align the findings of mediation-based studies on adults ([28]; [46]; [79]; [114]; [124]) and in one adolescent sample ([31]) and previous research among adolescents that has investigated these two paths separately ([16]; [28]; [44]; [43]; [53]; [60]; [73]; [74]; [84]; [91]; [94]).

As previously pointed out, physical activity participation may provide access to multiple sources of GSE ([6], [7]; [85]; [108]), including experiences of mastery, e.g., related to improved sports techniques and the acquisition of general skills like working systematically toward a goal ([20]; [79]; [92]). In addition, physical activity in social settings may facilitate the development of social skills, such as collaboration and leadership, and offer access to peers and adults who can serve as successful role models and provide support and encouragement ([36]; [83]; [88]; [125]). Furthermore, it is plausible that other identified mechanisms for the beneficial mental health effect of physical activity—both psychosocial, neurobiological and behavioural—also work indirectly through strengthened GSE beliefs. For instance, it is likely that improved cognitive capacity ([17]; [58]), better sleep ([42]; [58]), enhanced mood, reduced stress ([30]; [58]; [67]; [122]), and increased physical capacity ([17]; [58]) gained through physical activity can boost adolescents’ sense of having the cognitive, mental and physical resources necessary to cope with life’s challenges, i.e., greater GSE.

Subsequently, when adolescents’ GSE is strengthened through these various processes, it is suggested to improve well-being through enhanced optimism and greater engagement in societal activities ([5]; [10]; [68]; [85]; [100]). Our findings support this pathway, as the mediation model’s path b demonstrated a significant positive effect of GSE on SWB. Furthermore, positive spirals likely exist among factors inherent in the indirect pathway of our model. For instance, heightened participation in societal activities resulting from stronger GSE beliefs may offer additional opportunities for mastery experiences and supportive significant others ([10]; [68]), further enhancing GSE.

This study also found that the mediating role of GSE was more pronounced for girls and younger adolescents. There were no gender or age differences in the effect of physical activity on GSE; hence, these differential indirect effects were solely attributed to gender- and age-based variations in the effect of GSE on SWB. As previously discussed in a study applying a nearly identical sample of adolescents ([64]), the greater impact of GSE on girls’ quality of life may be due to their generally lower belief in their ability to handle challenging situations ([10]; [75]), likely related to their higher perceived stress ([73]) and pressure ([3]). Therefore, gaining a general sense of control may play a more critical role in girls’ well-being than in boys’ well-being. Additionally, girls’ greater internalisation of achievement values ([69]) and their higher potential for improvement, given their lower SWB ([15]; [44]; [73]), may contribute to explaining their stronger response to GSE.

The greater influence of GSE on the well-being of younger adolescents compared to older ones may be linked to their developmental phase, characterised by significant and rapid biological, cognitive, psychological, and social changes ([23]; [33]; [38]). These changes include pubertal development, development in self-concept, identity exploration, establishing autonomy from parents, transition to secondary school, heightened social comparison, and attempts to fit in with peer groups. This challenging stage may explain the lower GSE beliefs observed among younger adolescents in this study and others ([43]), and consequently, improvements in GSE may be particularly impactful for their well-being.

### 4.3. Practical Implications

Overall, our finding of a positive relationship between physical activity and SWB supports a continued focus on maintaining and promoting adolescent physical activity levels to help counter the decline in their mental health ([8]; [17]; [47]; [57]; [94]; [102]; [116]). This focus is particularly relevant given the extensive research demonstrating low physical activity levels among large segments of the adolescent population, both in Norway ([104]) and other Western countries ([109]). The notably lower levels of well-being and physical activity reported by girls and older adolescents, evident in this study and others ([2], [3]; [15]; [49]; [101]; [102]; [104]), highlight the need for particular attention to these sub-groups in physical activity-promoting public health initiatives.

Our study suggests that adolescents of both genders and all ages would benefit from increased physical activity levels in terms of improved SWB. However, findings of increasing direct effects with age and of greater indirect effects among younger adolescents and girls indicate that older adolescents may derive greater benefits from effects unrelated to GSE, while younger adolescents and girls particularly gain from physical activities that successfully strengthen their confidence in their abilities. To effectively strengthen GSE through physical activity, beyond potential effects from mere participation (via neurobiological, psychosocial and behavioural mechanisms that might enhance sense of cognitive, mental and physical capacities) ([13]; [30]; [31]; [42]; [58]; [67]; [122]), it seems crucial to facilitate personal mastery experiences within the physical activity setting, as these are considered the most powerful sources for enhanced self-efficacy ([6], [7]; [85]; [108]). This may involve improvements in specific aspects of physical activities, such as technical or strategic skills, and in general skills relevant to the activity setting, like social skills. Other key aspects of GSE-enhancing environments are access to significant others—such as leaders, coaches, parents, teachers, teammates, classmates, and friends—who serve as role models demonstrating that mastery and development are achievable and who assure the adolescents that they themselves possess the necessary resources to succeed with the various challenging tasks and situations they encounter ([10]; [85]; [108]; [113]). This would be relevant across the different arenas where adolescents engage in physical activities, including schools, sports clubs, and fitness centres. As demonstrated, it would be particularly beneficial in contexts involving younger adolescents and girls. Moreover, physical activity environments that foster mastery experiences and social support are anticipated to enhance intrinsic motivation for participation, thereby increasing the likelihood of sustained involvement in and further benefits from physical activity ([27]).

Achieving success in developing physical activity environments that effectively enhance adolescents’ GSE may require systemic efforts guided by a socio-ecological approach ([35]; [106], [107]). At the policy level, our results emphasise the importance of incorporating self-efficacy-building physical activity strategies into public health and education frameworks. This could include developing school and sports club programmes that prioritise improving GSE, supported by educational initiatives targeting sports club leaders, coaches, school leaders, teachers and parents. These initiatives should raise awareness about the concerning trends in adolescents’ mental health, the significance of strengthening GSE, and the role of physical activity in fostering these beliefs. It would be vital to communicate their key role in enhancing adolescents’ GSE by providing physical activity environments that support mastery experiences and emphasise positive role modelling and encouragement. Moreover, establishing sustainable mental health-promoting physical activity settings for adolescents is expected to be facilitated through formal collaboration among relevant local sectors and stakeholders, including local authorities, the school sector, health services, NGOs and the adolescents themselves ([32]).

### 4.4. Strengths and Limitations of the Study and Future Research

Drawing on data from a large sample of adolescents from the general population, this study provides novel empirical insights into the pathways through which physical activity impacts adolescents’ mental health, and how these effects vary among different population sub-groups. This knowledge advances theory in the field and forms the basis for better-targeted physical activity interventions aimed at improving the quality of life of this part of the population.

However, this study has several limitations. Primarily, the cross-sectional design challenges the ability to firmly conclude about causality and directions of effect. For instance, adolescents with higher well-being may be more likely to engage in physical activity. Longitudinal or experimental studies would provide more robust evidence regarding the direction of effects. Furthermore, the study focused solely on one potential mediator of physical activity’s effect on SWB, simplifying a complex array of mechanisms ([67]). Future studies should include multiple mediators (e.g., self-esteem, sleep patterns, social support) into serial and parallel mediation models ([51]). Moreover, aspects of physical activity behaviour other than frequency, such as intensity, type, variety and duration, may play a significant role in the development of GSE ([11]; [94]; [109]). However, these dimensions were not explored in our study and should be addressed in future research.

Additionally, while self-reporting physical activity is the most feasible method for large-scale studies as ours, it poses challenges in precision due to the complexity of recalling this behaviour and social desirability bias ([52]), potentially influencing the study’s results. Future research would benefit from combining self-report with objective, device-based assessment of physical activity ([109]). In addition, the generalisability of findings to the broader Norwegian adolescent population may be somewhat limited due to the higher attrition rates, due to missing data, among adolescents with lower levels of physical activity, GSE and SWB. Furthermore, pupils who did not consent to participate, were absent during data collection, or had dropped out, may also systematically differ from those included in the study in ways that may have impacted the results. Future studies on this topic should aim to secure more representative samples of adolescents. Finally, data collection occurred during the COVID-19 pandemic, following an extended lock-down period, which likely influenced physical activity levels, GSE beliefs and SWB, possibly affecting the study’s outcomes.

## 5. Conclusions

This study adds to the existing body of knowledge on how physical activity enhances well-being among adolescents. Our findings suggest that this effect partly stems from a strengthened belief in the ability to handle challenging situations, likely occurring through various parallel and cascading pathways. The study further suggests that the direct effect of physical activity on SWB increases with age, while the indirect effect through heightened GSE is more pronounced among girls and younger adolescents, as they benefit more from increased GSE. Overall, these findings emphasise the need for a continued focus on physical activity-promoting initiatives in public health efforts aimed at counteracting the ongoing trend of diminishing mental health in the adolescent population, and in these interventions, pay particular attention to GSE-enhancing efforts. Such investment in adolescents’ mental health is crucial not only for the individual adolescents and their families today but also for their future, future generations, and societal development ([120]).

## Figures and Tables

**Figure 1 behavsci-15-01239-f001:**
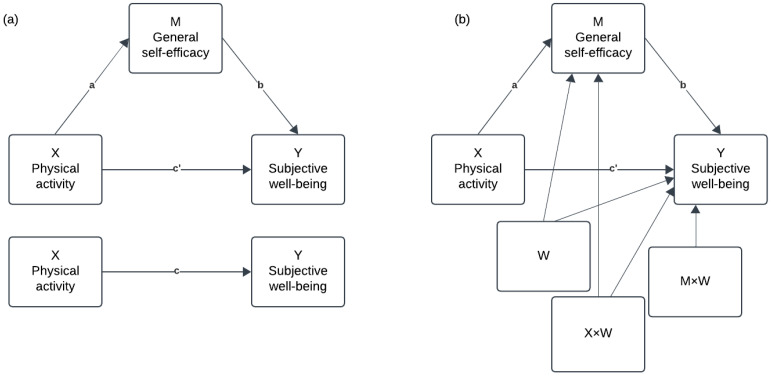
Simple mediation (**a**) and moderated mediation analysis (**b**). Note. X = independent variable, Y = dependent variable, M = mediating variable, W = moderating variable, a = effect of X on M, b = effect of M on Y, c’ = direct effect of X on Y, c = total effect of X on Y.

**Figure 2 behavsci-15-01239-f002:**
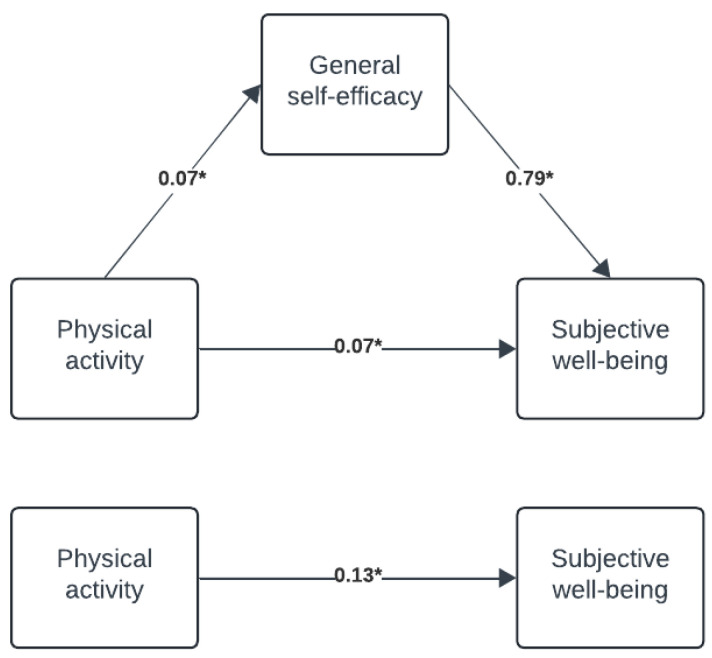
Mediation analysis. Note. Given in unstandardised regression coefficients and * = *p* < 0.001.

**Figure 3 behavsci-15-01239-f003:**
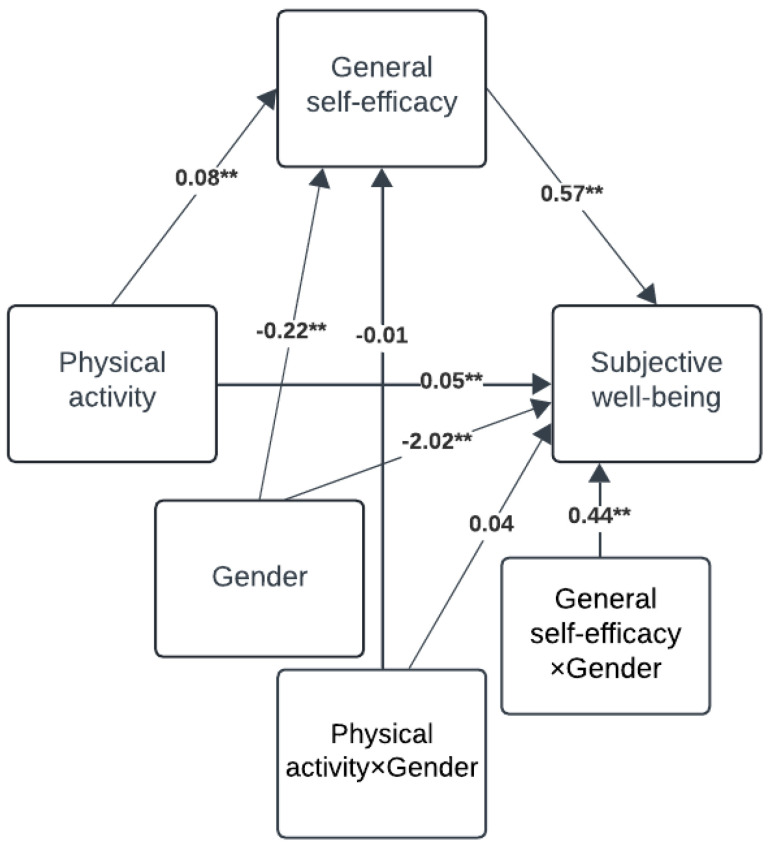
Moderated mediation analysis with gender as moderator. Note. Given in unstandardised regression coefficients and ** = *p* < 0.001.

**Figure 4 behavsci-15-01239-f004:**
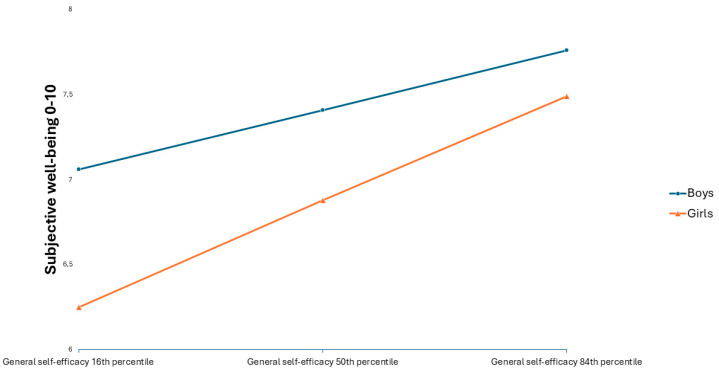
Interaction between general self-efficacy and gender on subjective well-being.

**Figure 5 behavsci-15-01239-f005:**
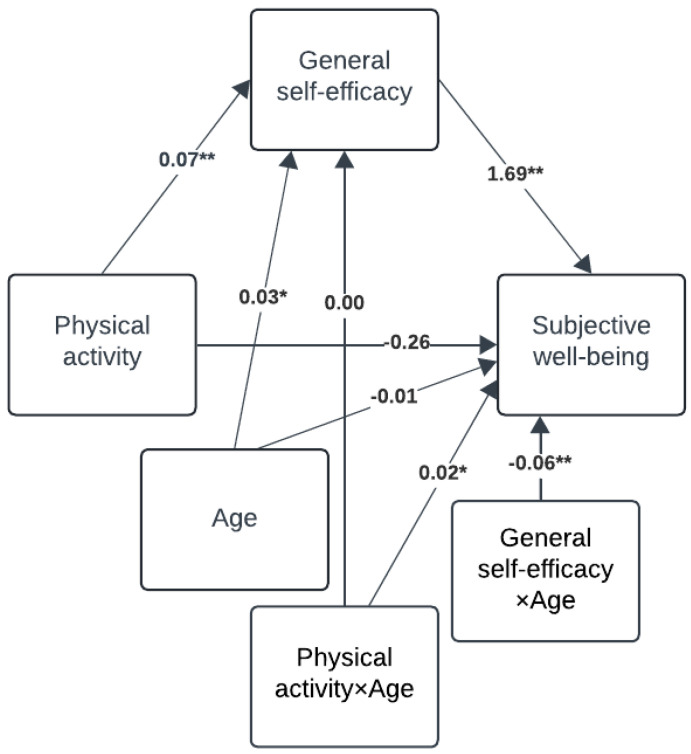
Moderated mediation analysis with age as moderator. Note. Given in unstandardised regression coefficients and * = *p* < 0.01, ** = *p* < 0.001.

**Figure 6 behavsci-15-01239-f006:**
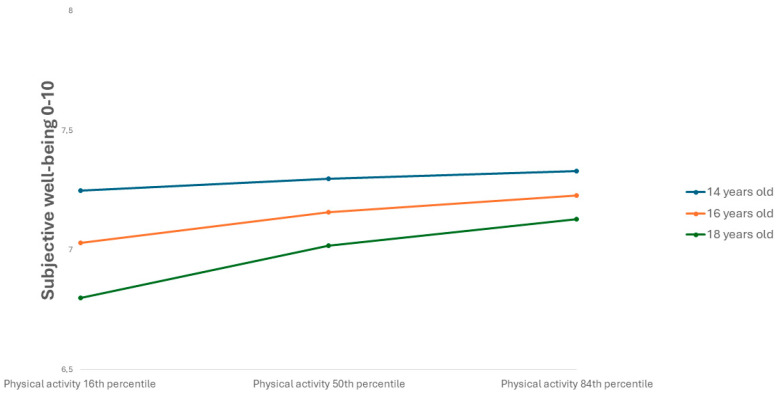
Interaction between physical activity and age on subjective well-being.

**Figure 7 behavsci-15-01239-f007:**
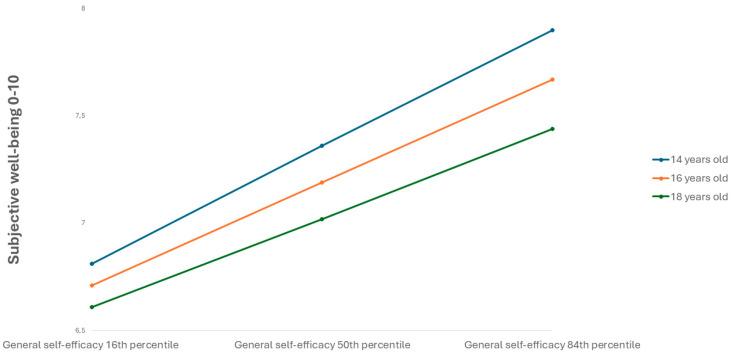
Interaction between general self-efficacy and age on subjective well-being.

**Table 1 behavsci-15-01239-t001:** Attrition analysis for the study.

Variables	Not Included in Study Sample			Included in Study Sample (*n* = 18,146)		M Diff	95 % CI M Diff	
	*n*	*M*	*SD*	*M*	*SD*			
Physical activity	2680	4.33	1.40	4.54	1.23	−0.21	−0.27	−0.16
General self-efficacy	1125	2.82	0.71	2.93	0.62	−0.11	−0.15	−0.07
Subjective well-being	3446	6.83	2.32	7.10	1.92	−0.28	−0.36	−0.19
Age	3223	3.19	1.62	3.30	1.61	−0.12	−0.18	−0.06
Socio-economic status	3569	1.89	0.57	2.02	0.53	−0.13	−0.15	−0.11
Social support	2799	1.60	0.67	1.73	0.57	−0.12	−0.15	−0.10
		*n*	%	*n*	%	Chi-square results		
Gender								
Boys		1994	58.1	8782	48.4			
Girls		1440	41.9	9364	51.6	X^2^(1, 22,028) = 108.01, *p* < 0.001		

Note. *M* = mean, *SD* = standard deviation, M diff = difference between means, CI = confidence interval.

**Table 2 behavsci-15-01239-t002:** Descriptive statistics for the total sample segmented by gender and age groups.

Variables	Total Sample *n* = 18,146	Boys *n* = 8782	Girls *n* = 9364	M Diff. [95% CI]	14–16 years *n* = 9835	17–19 years *n* = 8311	M Diff. [95% CI]
	*M* ± *SD*/*n* (%)	*M* ± *SD*/*n* (%)	*M* ± *SD*/*n* (%)		*M* ± *SD*/*n* (%)	*M* ± *SD*/*n* (%)	
Subjective well-being 0–10	7.10 ± 1.92	7.52 ± 1.78	6.71 ± 1.96	−0.81 [−0.87, −0.76]	7.16 ± 1.97	7.03 ± 1.86	−0.13 [−0.18, −0.07]
Physical activity 1–6	4.54 ± 1.23	4.69 ± 1.24	4.40 ± 1.22	−0.15 [−0.33, −0.25]	4.59 ± 1.21	4.48 ± 1.26	−0.11 [−0.15, −0.08]
General self-efficacy 1–4	2.93 ± 0.62	3.07 ± 0.61	2.79 ± 0.59	−0.28 [−0.29, −0.26]	2.89 ± 0.63	2.97 ± 0.60	0.08 [0.05, 0.09]
Gender							
Boys	8782 (48.4)				4857 (49.4)	3925 (47.2)	
Girls	9364 (51.6)				4978 (50.6)	4386 (52.8) *	
Age 14–19	16.30 ± 1.61	16.24 ± 1.58	16.36 ± 1.64	0.12 [0.07, 0.16]			
Socio-economic status 0–3	2.02 ± 0.53	2.00 ± 0.52	2.03 ± 0.54	0.03 [0.02, 0.05]	2.05 ± 0.51	1.97 ± 0.55	−0.08 [−0.09, −0.06]
Social support 0–2	1.73 ± 0.57	1.75 ± 0.56	1.71 ± 0.58	−0.04 [−0.05, −0.02]	1.69 ± 0.59	1.77 ± 0.54	0.08 [0.05, 0.09]
No	1137 (6.3)	522 (5.9)	615 (6.6)		686 (7.0)	451 (5.4)	
Don’t know	2677 (14.8)	1184 (13.5)	1493 (16.0)		1631 (16.6)	1046 (12.6)	
Yes	14,332 (79)	7076 (80.6)	7256 (77.5)		7518 (76.4)	6814 (82.0)	

Note. *M* = Mean, *SD* = standard deviation, M diff. = difference between means, CI = confidence interval. Higher scores on continuous/ordinal variables reflect higher levels of the given indicator. Differences between genders and age groups are tested by *t*-tests for continuous variables and Chi-square-tests for categorical variables. * X^2^(1, 18,146) = 8.402, *p* = 0.004.

**Table 3 behavsci-15-01239-t003:** Correlations among study variables for the total sample (*n* = 18,146).

Variables	1	2	3	4	5	6
1. Subjective well-being						
2. Physical activity	0.16 *					
3. General self-efficacy	0.35 *	0.20 *				
4. Gender	−0.21 *	−0.12 *	−0.23 *			
5. Age	−0.04 *	−0.04 *	0.07 *	0.04 *		
6. Socio-economic status	0.10 *	0.21 *	0.10 *	0.03 *	−0.08 *	
7. Social support	0.30 *	0.10 *	0.22 *	−0.04 *	0.07 *	0.08 *

Note. Boys = 0, Girls = 1. Higher scores on continuous/ordinal variables reflect higher levels of the given indicator. Correlations with physical activity and social support are based on Spearman Rank Order correlation analyses, otherwise Pearson correlations are presented (* = *p* < 0.001).

**Table 4 behavsci-15-01239-t004:** Correlations among study variables by gender.

	Variables	1	2	3	4	5
Boys (*n* = 8782)	1. Subjective well-being					
	2. Physical activity	0.12 *				
	3. General self-efficacy	0.25 *	0.19 *			
	4. Age	−0.10 *	−0.01 *	0.09 *		
	5. Socio-economic status	0.09 *	0.19 *	0.11 *	−0.06 *	
	6. Social support	0.27 *	0.11 *	0.19 *	0.01 *	0.09 *
Girls (*n* = 9364)	1. Subjective well-being					
	2. Physical activity	0.15 *				
	3. General self-efficacy	0.38 *	0.18 *			
	4. Age	−0.01 *	−0.06 *	0.07 *		
	5. Socio-economic status	0.12 *	0.24 *	0.11 *	−0.09 *	
	6. Social support	0.32 *	0.09 *	0.25 *	0.13 *	0.07 *

Note. Boys = 0, Girls = 1. Higher scores on continuous/ordinal variables reflect higher levels of the given indicator. Correlations with physical activity and social support are based on Spearman Rank Order correlation analyses, otherwise Pearson correlations are presented (* = *p* < 0.001).

**Table 5 behavsci-15-01239-t005:** Correlations among study variables by age groups.

	Variables	1	2	3	4	5
14–16 years (*n* = 9835)	1. Subjective well-being					
	2. Physical activity	0.13 *				
	3. General self-efficacy	0.38 *	0.19 *			
	4. Gender	−0.25 *	−0.11 *	−0.22 *		
	5. Socio-economic status	0.08 *	0.21 *	0.09 *	0.05 *	
	6. Social support	0.32 *	0.14 *	0.24 *	−0.08 *	0.08 *
17–19 years (*n* = 8311)	1. Subjective well-being					
	2. Physical activity	0.18 *				
	3. General self-efficacy	0.31 *	0.22 *			
	4. Gender	−0.17 *	−0.13 *	−0.23 *		
	5. Socio-economic status	0.11 *	0.21 *	0.12 *	0.02	
	6. Social support	0.28 *	0.11 *	0.19 *	0.02	0.09 *

Note. Boys = 0, Girls = 1. Higher scores on continuous/ordinal variables reflect higher levels of the given indicator. Correlations with physical activity and social support are based on Spearman Rank Order correlation analyses, otherwise Pearson correlations are presented (* = *p* < 0.001).

## Data Availability

The data that support the findings of this study are available for researchers upon request to the [81] ([81]).

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
