# Peer review of "General Self-Efficacy as a Mediator of Physical Activity’s Impact on Well-Being Among Norwegian Adolescents: A Gender and Age Perspective"

_behavsci, 2025, doi:10.3390/bs15091239_

Round 1

Reviewer 1 Report

Comments and Suggestions for Authors

It is the interesting research and do help us to deepen the understanding of the mechanism by which PA influences SWB. However, there are some issues have to be clarified before publication consideration.

  1. line 47-49. After 2020, some review papers addressed on the influencing mechanisms between PA and mental health, as well as well-being, were published. However, this study has not referred to any of them. Such as, (1)White et al.Physical activity and mental health:a systematic review and best-evidence synthesis of mediation and moderation studies. Int J Behav Nutr Phys Act (2024) 21:134. https://doi.org/10.1186/s12966-024-01676-6. (2)Marquez et al.  A systematic review of physical activity and quality of life and well-being. Translational Behavioral Medicine, Volume 10, Issue 5, October 2020, Pages 1098–1109, https://doi.org/10.1093/tbm/ibz198.
  2. The dependent variable of the study is subjective well-being. Why do the literature reviews keep talking about mental health instead of directly addressing the main topic?
  3. Previous studies related to PA and subjective well-being were also not explained in the introduction section, which resulted in the rational explanation of the research being unclear and not direct enough.
  4. line 203-208. As an independent variable, PA is strictly a categorical variable because its classification method in your study. And also, the classificaton method in this study is different from the methods used by WHO or ACSM to define the frequency of PA. How can it be regarded as an ordinal variable? Should it be analyzed as a continuous variable instead?
  5. About the moderated mediation analysis (b) in figure 1, please present the process of model or path selection in a table.

Reviewer 2 Report

Comments and Suggestions for Authors

Theoretical contribution and contextualisation:

The article is well grounded theoretically and offers an extensive review of previous literature, citing recent and relevant studies (e.g., Biddle et al., 2019; Lubans et al., 2016). However, some sections of the theoretical framework present redundant ideas. It would be useful to better synthesise the sections describing the neurobiological and psychosocial mechanisms to give greater weight to the justification of the proposed model (especially the relationships between physical activity, general self-efficacy (GSE) and subjective well-being).

Structure and clarity of arguments:

The logic of the study is well outlined, but in several sections (e.g., in the Discussion, lines 434–503) the arguments are repeated with minimal variations. We suggest carefully reviewing these paragraphs and consolidating the key arguments to avoid redundancy. This will not only make the text clearer, but also make it easier to read the main findings.

Results and Tables:

The tables (especially Table 2 and Table 3) are comprehensive but very dense. We recommend:

Highlighting or summarising the most statistically relevant findings in the text (e.g., those correlations with the largest effect sizes).

Consider a more visual or simplified format for long tables, using summaries or graphs (figures) where possible.

Include in the text a briefer interpretation of the effect sizes (small, medium, large) according to Cohen (1988) to guide the reader.

Discussion and practical contributions:

The section on practical implications (lines 506–541) is solid, but could link the recommendations more closely to the statistical results. For example, explain how the size of the mediating effect of GSE in girls and young adolescents suggests differentiated programmes for these groups. It would also be interesting to discuss the policy implications (e.g., designing school programmes that integrate strategies to improve self-efficacy).

Limitations and future research:

Although the limitations of the cross-sectional design are addressed (lines 561–573), it is recommended to emphasise the risk of reverse causality (e.g., adolescents with higher well-being may be more physically active). In addition, it would be valuable to suggest longitudinal or experimental models to corroborate the direction of the effects, as well as to include multiple mediators (e.g., self-esteem, sleep quality, or social interactions), following multiple mediation models.

References:

There are references marked “Author, under review” (e.g., lines 75, 132, 145, 485). It is important to update these before publication to ensure the traceability of sources.

Originality and relevance:

The study provides robust data with a large sample (n = 18,146) and addresses a significant gap in the literature on the mediating role of general self-efficacy in adolescents. This gives it high scientific value and relevance to the field of public health and youth well-being.

Comments on the Quality of English Language

English quality:

In general, the English is correct, but there is a tendency to use very long sentences loaded with subordinate clauses, which makes for difficult reading.

Example: “Consequently, acquiring a general sense of control is likely more important for girls’ well-being compared to boys.” (line 489). This type of sentence could be simplified to: ‘Therefore, gaining a general sense of control may play a more critical role in girls' well-being than in boys'.’

There is frequent use of passive constructions, which detracts from clarity. A more direct and active style would make the text more dynamic.

A professional review of the English language is recommended, especially to improve conciseness and punctuation.
